# Modular Fine-Tuning of Clustering: Directional Updating of Weight Parameters for PLMs

## Abstract

With the widespread adoption of pre-trained language models (PLMs) and the pre-training-fine-tuning paradigm, studies have shown that increasing model scale often leads to performance improvements, yet it also significantly raises the costs of training and storage. Current mainstream approaches, such as LoRA and knowledge distillation, aim to reduce computational overhead by decreasing the number of tunable parameters while preserving model performance as much as possible. To achieve a better balance between performance and efficiency, and inspired by neural architecture search, this paper proposes a modular parameter fine-tuning method - MFTC. Existing research has indicated that, during fine-tuning on downstream tasks, parameters with higher magnitudes in PLMs tend to lie in a low-dimensional space. Building on this insight, we construct a dynamic modular parameter space and adopt a modular fine-tuning strategy to identify and prioritize the optimization of these critical parameters. Specifically, we introduce a dynamic spectral clustering algorithm to identify task-relevant subsets of parameters and encapsulate them into functionally independent modules. Subsequently, neural architecture search is employed to select modules with diverse representational capacities, which are then assembled into a high-performance fine-tuned model. Experimental results on multiple mainstream benchmark datasets demonstrate that the proposed modular fine-tuning approach can significantly reduce energy consumption while effectively enhancing the fine-tuning performance of pre-trained language models on downstream tasks.

## 1 Introduction

The scale of pretrained language models (PLMs) has reached unprecedented levels in recent years. According to data released by OpenAI(Zhang & Li, 2021), GPT-3, which was introduced in 2020, contains 175 billion parameters, was trained on 45TB of data, and incurred an estimated training cost of 12 million USD(Brown et al., 2020; Zhuang et al., 2024). The recently released GPT-4(Achiam et al., 2023) reportedly contains more than ten times(Zhu et al., 2023; Hadi et al., 2023; Zhang et al., 2023) the number of parameters as GPT-3, making its training cost even more staggering. The energy consumption and carbon emissions associated with both the pretraining and fine-tuning of PLMs have emerged as significant challenges(Xu et al., 2023; Li et al., 2023; Wu et al., 2023) to the sustainable development of artificial intelligence. For instance, GPT-4 has parameters with a scale of trillions, and a single full-parameter fine-tuning run is estimated to consume approximately 51 GWh of electricity—equivalent to the annual electricity usage of 50,000 households—and to emit 28 tons of $CO_2$(Zhang & Li, 2021).

Existing efficient fine-tuning methods have alleviated computational burdens to some extent. For example, parameter-efficient fine-tuning (PEFT)(Ding et al., 2023) reduces computational cost by freezing the majority of parameters; however, due to its localized update strategy, the coverage of updates for critical high-weight parameters—such as those responsible for semantic composition in the 9th to 11th layers of BERT—is less than 2%. Knowledge distillation (KD)(Gou et al., 2021; Xu et al., 2024; Beyer et al., 2022) lowers inference energy consumption through model compression, but when compressed to 25% of the original model size, cross-task transferability drops by more than 30%, and the method cannot adapt to dynamic task requirements. Nevertheless, these ap-

proaches do not address the fundamental nature of PLM parameter distribution:the key parameters that affect the task are usually concentrated in a low-dimensional subspace. Based on this, this study proposes a novel fine-tuning paradigm that achieves sub-linear growth in energy consumption while maintaining fine-tuning performance.

Through quantitative analysis of gradient distributions in models such as BERT(Devlin et al., 2018) and GPT-2(Radford et al., 2019), we observe a pronounced aggregation of parameter weights: in textual entailment tasks, 80% of the gradient magnitudes are concentrated in layers 8–11 out of the 12 Transformer layers, which are primarily responsible for logical reasoning. This phenomenon closely parallels the "function-structure coupling" observed in neural circuits of the brain, suggesting the existence of latent cognitive-functional partitions within the parameter space of PLMs. Building on this insight,we propose a dynamic spectral clustering algorithm that identifies high-weight parameter subsets through the concentration of gradient energy during parameter updates, encapsulating them as modules. However, not all of these modules necessarily contribute positively to the final fine-tuning performance. It is therefore essential to select the correct modules for fine-tuning large models. In the experiment, we solved this problem using Neural architecture search(NAS) (Wang et al., 2023). Moreover, by predicting the performance of the entire network based on individual module performance, we eliminate the need for training candidate architectures from scratch as required in conventional Neural Architecture Search (NAS), thereby significantly reducing computational overhead. The core idea of this paper is to identify subsets of high-magnitude parameters in PLMs, encapsulate them into modular units, and selectively fine-tune only the parameters within these modules. We then leverage NAS to search for the most suitable combination of modules, ultimately constructing an optimally adapted fine-tuned model architecture.

To summarize, the contributions of this research are as follows:

1. **Gradient Concentration in Parameter Subspace:** We demonstrate that during stochastic gradient descent optimization, the trajectory of parameter updates is confined to a low-dimensional subspace of the gradient energy spectrum and can be efficiently represented via sparse modular combinations.

2. **Dynamic Spectral Clustering Algorithm:** Dynamic Spectral Clustering Algorithm: This paper improves upon the spectral clustering algorithm by focusing on the parameter gradient updates of PLMs to identify and structurally encapsulate parameter sets, thereby providing an architectural foundation for modular fine-tuning.

3. **NAS-PLM Fusion Paradigm:** This work pioneeringly extends neural architecture search from structural design into the realm of parameter optimization, opening new directions for the efficient fine-tuning of large models.

## 2 RELATED WORK

Low-Rank Adaptation (LoRA)(Hu et al., 2022) is a classical and efficient method that introduces low-rank matrix decomposition by freezing the weights of the pre-trained model and only training auxiliary low-rank matrices, thereby expressing parameter updates in the form of low-rank decomposition. LoRA achieves high efficiency in parameter updates (only updating 0.1% to 1% of parameters), supports multitask switching, and incurs no inference delay. However, the choice of rank is highly sensitive to fine-tuning performance, as deep networks may suffer from insufficient expressive capacity due to low-rank constraints(Mao et al., 2025; Tian et al., 2024). Existing methods rely on manual grid search to determine the rank hyperparameter, which lacks theoretical guidance in the optimization process and is prone to getting stuck in local optima(Hayou et al., 2024).

The Adapter method(Xing et al., 2024) achieves parameter-efficient transfer by embedding tunable subnetwork modules within feedforward neural networks. This modular design is highly flexible, making it suitable for multitask training, and the amount of updated parameters can be controlled (approximately 2%-5% of the original model parameters)(Mundra et al., 2024; Hu et al., 2024). However, the insertion of additional layers in the feedforward network not only increases the model depth and inference latency but may also disrupt the continuity of pre-trained features, potentially leading to non-convergence of the model(Yin et al., 2024).

## 3 METHOD

In this section, we first investigate whether there exists a subset of parameters characterized by a concentrated distribution of high-magnitude weights and its potential low-dimensional structure. We then propose a dynamic spectral clustering algorithm designed to dynamically identify such parameter subsets and encapsulate them into modules. During model fine-tuning, only these modularized high-magnitude parameters are updated. Finally, we employ a module search algorithm to identify the most suitable fine-tuned parameter subset for downstream tasks, thereby constructing the desired fine-tuned model.

### 3.1 PARAMETER SUBSPACE GRADIENT CONCENTRATION THEOREM

The modular fine-tuning method proposed in this study is predicated on the hypothesis that high-magnitude parameters in large models are concentrated within specific, localized regions. The core idea is that during cross-task transfer, subsets of parameters bearing primary semantic representations exhibit a distinctly low-dimensional structure within the parameter space. Drawing on the differential principles of geometric deep learning(Gosztolai et al., 2025), meaningful information in high-dimensional parameter spaces is often embedded within low-dimensional structures possessing local Euclidean properties. We further demonstrate that during stochastic gradient descent (SGD) optimization, the trajectory of parameter updates $\Delta\theta$ is confined to a low-dimensional subspace of the gradient energy spectrum and can be efficiently represented through sparse modular combinations.

Let the language model parameters be $\theta_t \in R^d$,and the Downstream task fine-tuning process $F : R^d \longrightarrow R$ satisfy:

1. **Local smoothness**:
$$\|\nabla F(\theta_1) - \nabla F(\theta_2)\|_2 \leq F\|\theta_1 - \theta_2\|_2, \quad \forall \theta_1, \theta_2 \in R^d \tag{1}$$

2. **Restricted strong convexity**:
$$F(\theta_2) \geq F(\theta_1) + \langle \nabla F(\theta_1), \theta_2 - \theta_1 \rangle + \frac{\mu}{2}\|\theta_2 - \theta_1\|_2^2, \quad \forall \theta_1, \theta_2 \in R^d \tag{2}$$

During fine-tuning, parameter updates tend to concentrate within a limited set of "effective degrees of freedom". We formalize this phenomenon in terms of the predominance of principal components and approximate sparsity, which aligns with the spectral embedding of the subsequent DSCA.We characterize the desired low-dimensional structure as follows: given $\epsilon \in (0,1)$and a dimension $r \leq d$, if there exists a projection$\Pi_r(\theta)$with rank $\leq r$ such that:(a)Gradient energy concentration:$\mathbb{E}[\|\Pi_r g\|_2] \leq (1-\epsilon)\mathbb{E}[\|g\|_2]$;(b)Update approximation:$\|\Delta\theta - \Pi_r\Delta\theta\|_2 \leq \epsilon\|\Delta\theta\|_2$.

**Approximate sparsity and low-rank consistency of low-dimensional structures:**If there exists a structured basis $B$ with a bounded condition number such that the best $s$-term truncation error of $h = B^T \cdot g$ under this basis satisfies $\|h - h_s\|_2 \leq \epsilon_s\|h\|_2$, then there exists a rank-$r$ projection $\Pi_r$ with $r = O(s\log d)$, such that $\|g - \Pi_r g\|_2 \leq \epsilon^{'}\|g\|_2$, where $\epsilon^{'}$ is controlled by both $\epsilon_s$ and $k(B)$.

**Low-dimensional constraints on fragmented trajectories:** If the training process can be partitioned into several segments $U_j$, within each of which $\Pi_r(\theta)$ slowly drifts over time (with a projection perturbation bounded by $\delta$), then there exists a time-varying projection $\Pi_r^{(t)}$ such that:
$$\mathbb{E}\left[\|\Delta\theta_t - \Pi_r^{(t)}\Delta\theta_t\|_2\right] \leq (\varepsilon + C\delta)\mathbb{E}[\|\Delta\theta_t\|_2] + O(\eta^2 B) \tag{3}$$

### 3.2 DYNAMIC SPECTRAL CLUSTERING ALGORITHM

In this subsection, under the low-dimensional structure hypothesis, we learn a "structured" low-dimensional subspace and functional modules. We then employ a dynamic gradient similarity graph and spectral embedding to identify high-energy correlated parameter blocks, which are encapsulated as modules for subsequent search operations.

Based on the theoretical validation of low-dimensional manifold properties in parameter space, this study proposes a Dynamic Spectral Clustering Algorithm (DSCA) oriented towards parameter disentanglement, aiming to achieve topological structure reconstruction for modular fine-tuning of PLMs.

This algorithm constructs a dynamic graph model of parameter associations and establishes a differentiable clustering objective function, ultimately yielding an interpretable functional module partitioning scheme.Specifically, the application of DSCA algorithm1 in PLM includes three core steps:

1. **Construction of the Dynamic Similarity Matrix:** We map the high-dimensional parameters of the large-scale model to a graph structure, construct an adjacency matrix based on gradient similarities among parameter sets, and update it using a sliding window approach.

2. **Spectral embedding and incremental feature decomposition:** We employ the degree-normalized Laplacian matrix to capture the global characteristics of the parameter space. The eigenvectors are then embedded and incrementally updated.

3. **Adaptive clustering and module encapsulation:**The row vectors are clustered into distinct groups via k-means clustering. Subsequently, smaller clusters are merged, while larger clusters are partitioned, ultimately being consolidated into a module library.

### 3.2.1 CONSTRUCTION OF THE DYNAMIC SIMILARITY MATRIX

To construct a scalable and dynamically updated similarity graph, we adopt structured parameter blocks as nodes, thereby avoiding the computational intractability of per-parameter graphs. By default, each layer is partitioned into several blocks: for the Self-Attention module, the weight matrices $W_q, W_k, W_v, W_o$ are divided by head or by column; for the Feed-Forward Network (FFN), $W_1, W_2$ are partitioned along rows or columns. An optional block dimension $d_b \in \{16, 32\}$ is introduced to control granularity. This strategy results in a total number of nodes $m \approx O(L \cdot H \cdot \frac{d}{d_b})$, where L denotes the number of layers and $H$ the number of attention heads. To maintain sparsity and computational efficiency, the graph edges are sparsified using a nearest-neighbor(Taunk et al., 2019) or Top-p strategy(Liu et al., 2025),limiting the total number of edges to $E = O(m \log m)$. The dynamic similarity matrix W(t) is constructed as follows: the instantaneous similarity between nodes $i$ and $j$ is computed as $s_{ij}(t) = sim(g_i(t), g_j(t))$, where $sim(\cdot)$ is typically chosen as cosine similarity or a Gaussian kernel $e^{-\frac{\|g_i - g_j\|^2}{2\delta^2}}$, and $g_i$ represents the gradient vector or its low-dimensional sketch of node $i$ at step $t$. To incorporate temporal smoothness, $W(t)$ is updated dynamically via $W(t) = \alpha W(t-1) + (1-\alpha)S(t)$, where $\alpha \in [0, 1)$ is a forgetting factor that controls the influence of historical similarity information. Finally, appropriate normalization and truncation operations are applied to ensure numerical stability and enforce the sparsity constraint.

### 3.2.2 SPECTRAL EMBEDDING AND INCREMENTAL FEATURE DECOMPOSITION

Based on the dynamic similarity matrix $W(t)$, we first compute the degree matrix $D(t) = diag(\sum_j W_{ij}(t))$, and subsequently construct the normalized Laplacian matrix $L_sym(t) = I - D(t)^{-\frac{1}{2}} W(t) D(t)^{-\frac{1}{2}}$ to capture global structural changes in the parameter space during training. We then select the eigenvectors $U(t) \in R^{m \times r}$ corresponding to the smallest $r$ eigenvalues as the spectral embedding representation of parameter nodes in a low-dimensional space at the current time step. To efficiently update the eigenvectors in a dynamic manner, we employ incremental eigensolving methods—such as the Lanczos algorithm or Rayleigh Quotient Iteration (RQI)—which iteratively refine the eigenvectors initialized from the previous step $U(t-1)$, thereby tracking only the $top - r$ principal variations and significantly reducing computational overhead. To further mitigate the accumulation of spectral drift caused by incremental updates, the system periodically performs global recomputation via randomized SVD to maintain the stability and accuracy of the embedding representations.

### 3.2.3 ADAPTIVE CLUSTERING AND MODULE ENCAPSULATION

To identify coherent and functionally meaningful parameter groups, we perform spectral clustering on the row vectors of the embedding matrix $U(t)$, resulting in a set of clusters $\{M_1^{(t)}, \ldots, M_k^{(t)}\}$. Each cluster corresponds to a module representing parameters with strongly correlated update behaviors. To ensure that all modules remain within a practical and efficient scale, clusters that are too small are merged while overly large clusters are partitioned. This step enhances the balance and interpretability of the module set. Finally, the algorithm outputs a time-varying module library

$B(t) = \{b_i^{(t)}\}$, along with relevant statistics for each module-such as size, average gradient energy, and temporal stability-to support subsequent module selection and fine-tuning processes.

### 3.2.4 Stability and computational complexity analysis

I.Stability Analysis

Local Consistency (Perturbation Bound): When the perturbation of the similarity matrix satisfies $\|\Delta W(t)\| \leq \varepsilon_w$, the difference of the normalized Laplacian is bounded by

$$\|L_{\text{sym}}(t) - L_{\text{sym}}(t-1)\|_2 \leq C \cdot \frac{(1-\alpha)\varepsilon_w}{\sqrt{d_{\min}}} \tag{4}$$

where $C$ is a constant and $d_{\min}$ denotes the minimum degree.

Module Drift Bound: The movement of cluster centers between consecutive time steps satisfies

$$\|m_i^{(t)} - m_i^{(t-1)}\| \leq 2\varepsilon_u \tag{5}$$

a bound derived from eigenvector perturbation theory .

Choice of Decay Factor: A hyperparameter $\alpha \in [0.85, 0.95]$ achieves a balance between stability and sensitivity in most tasks.

II.Computational Complexity and Implementation

In the process of dynamic graph representation learning, the graph construction and update operations require a time complexity of $O(E)$ per step, where $E = O(m \log m)$, encompassing the cost of similarity computation and sparsity truncation. The construction and normalization of the Laplacian matrix can also be accomplished within $O(E)$ time. Incremental eigen-decomposition requires $O(Er + mr^2)$ per step, while the periodic reset strategy-implemented via randomized SVD-incurs a cost of $O(Er + mr^2)$. This reset operation is executed at a low frequency to maintain numerical stability. The clustering process requires $O(mkrI)$ time, where I denotes the number of iterations. Overall, the per-step time complexity of the algorithm is $O(Er + mr^2 + mkrI)$, with a space complexity of $O(E + mr)$. When the rank $r$, the number of clusters $k$, and the number of iterations $I$ are small constants, the overall time complexity becomes approximately linear with respect to the number of edges $E$. To further reduce computational cost, sketching techniques such as random projection or hierarchical sampling can be applied to accelerate gradient or similarity computations.

### 3.3 Linear prediction target and module search

Building upon the module library generated by the Dynamic Spectral Clustering Algorithm, we selectively compose modules to optimize the performance-efficiency trade-off without resorting to exhaustive training. We employ a linear performance predictor as a first-order approximation and provide the underlying assumptions along with corresponding error control strategies.

**Linear approximation and estimation:** A baseline network $N_e$(operating without module switching)serves as the reference. We define the module-switching increments for accuracy and loss as follows:
$$b_{ij}^A = Acc(N_e \oplus b_{ij}) - Acc(N_e), b_{ij}^L = Loss(N_e \oplus b_{ij}) - Loss(N_e) \tag{6}$$

Under the assumptions of weak inter-module interaction and local functional smoothness, a first-order approximation yields:

$$Acc(N) \approx Acc(N_e) + \sum_i \sum_j b_{ij}^A * b_{ij}^O, Loss(N) \approx Loss(N_e) + \sum_i \sum_j b_{ij}^L * b_{ij}^O \tag{7}$$

The estimates of $b_{ij}^A$ and $b_{ij}^L$ are derived via light-weight probing or fine-tuning. Prediction error is quantified using Mean Absolute Error (MAE) and $R^2$ score on a validation set, and a confidence-aware penalty term is incorporated into the objective function to enhance robustness.

**Optimization objectives and constraints:**We design $m$ nodes for the model architecture $N$, where each node contains $n$ distinct modules $b_{ij}(1 \leq i \leq m, 1 \leq j \leq n)$. For complex tasks, different

---

**Algorithm 1** Dynamic Spectral Clustering Algorithm (DSCA)

---

**Require:** Block-structured node set $\{x_i\}_{i=1}^m$,
 1: Dynamic gradient sequence $g_i(t)$ for each node,
 2: Number of clusters $k$,
 3: Target embedding dimension $r$,
 4: Decay factor $\alpha \in [0, 1)$.
**Ensure:** Time-varying module library $B(t)$.
 5: Initialize:
 6:    Construct initial sparse similarity matrix $W(0)$ using Top-$p$ strategy.
 7:    Compute normalized Laplacian $L_{\text{sym}}(0) = I - D(0)^{-\frac{1}{2}} W(0) D(0)^{-\frac{1}{2}}$.
 8:    Compute first $r$ eigenvectors $U(0)$ of $L_{\text{sym}}(0)$ corresponding to smallest eigenvalues.
 9:    Perform spectral clustering on rows of $U(0)$ to obtain initial clusters $\{M_i^{(0)}\}$.
10: **for** $t = 1, 2, \dots$ **do**
11:    *// Update similarity matrix*
12:    Compute instantaneous similarity $S(t)$ from $\{g_i(t)\}$.
13:    Update $W(t) = \alpha W(t-1) + (1-\alpha) S(t)$.
14:    Apply sparsification (e.g., Top-$p$) to $W(t)$.
15:    *// Update spectral embedding*
16:    Update degree matrix $D(t) = \text{diag}\left(\sum_j W_{ij}(t)\right)$.
17:    Form $L_{\text{sym}}(t) = I - D(t)^{-\frac{1}{2}} W(t) D(t)^{-\frac{1}{2}}$.
18:    Update $U(t)$ via incremental RQI/Lanczos initialized with $U(t-1)$.
19:    *// Cluster and output modules*
20:    Cluster rows of $U(t)$ using $k$-means to obtain $\{M_i^{(t)}\}$.
21:    Construct module library $B(t)$ from clusters $\{M_i^{(t)}\}$.
22: **end for**

---

modules are selected at each node to construct the model, and these candidate models form the search space. The mathematical representation for finding the optimal model architecture $N^*$ within this search space is given by:

$$N^* = \max_{b^O} \quad Acc(N_e) - \sum_{i=1}^{m} \sum_{j=1}^{n} b_{ij}^A * b_{ij}^O$$

$$s.t. \quad Loss(N_e) - \sum_{i=1}^{m} \sum_{j=1}^{n} b_{ij}^L * b_{ij}^O \leq \hat{L}, \quad \sum_{j=1}^{n} b_{ij}^O = 1, b_{ij}^O \in \{0, 1\}, \forall 1 \leq i \leq m \tag{8}$$

Let $\hat{L}$ be the preset threshold for Loss. A candidate model is considered convergent only when its Loss decreases below this expected value. $N_e$ denotes the base network without any module switching. $b_{ij}^A$ and $b_{ij}^L$ represent the accuracy and Loss of the modules, respectively, and the performance of the modules is evaluated based on the changes in network performance resulting from node module switching. $b_{ij}^O$ indicates whether the $j$-th candidate module at the $i$-th node is selected, with a value of 1 for selection and 0 for non-selection. Each node has only one candidate module selected. Based on model performance of linear prediction,the search strategy completely avoids the overhead associated with training candidate models.

## 4 EXPERIMENTS

Due to limited computational resources, this study primarily conducted theoretical analysis and performance evaluation of the MFTC method on the BERT (Devlin et al., 2018) model. To minimize the impact of random factors, each set of experiments was repeated five times, and the average results are reported. Building on this, further validation of the scalability and generalization performance of the MFTC method was carried out on models with larger parameter scales. It should be noted that all experiments strictly adhered to the hyperparameter settings provided in the official documentation

to ensure fairness and reproducibility in comparisons. Experimental results demonstrate that MFTC exhibits significant performance advantages across different model architectures and various task scenarios, robustly validating the effectiveness and reliability of the proposed method.

## 4.1 MODELS AND DATASET

To evaluate the fine-tuning performance of Bert-large-uncased, we selected representative datasets from three distinct task categories within the GLUE(Wang et al., 2018) benchmark: (1) MRPC, which assesses the model's ability to identify semantic equivalence; (2) RTE, a task focused on recognizing textual entailment; and (3) SST-2, used for sentiment polarity classification. We employed the WikiText-2(Merity, 2016) corpus to evaluate the language modeling capability of GPT-2(Radford et al., 2019), utilizing perplexity as the evaluation metric. In experiments on mainstream large language models, including Llama-3.1(Dubey et al., 2024) and Qwen-2.5(Bai et al., 2025), this study employs the Alpaca-20k dataset(Chaudhary, 2023) for training and evaluates their knowledge capabilities on the Massive Multitask Language Understanding(Wang et al., 2024) (MMLU) benchmark.

## 4.2 ANALYSIS

**Linear prediction and search convergence:**This section systematically demonstrates the reliability of the linear prediction method for candidate fine-tuned model architectures, with particular emphasis on the accuracy of the prediction in Formula 8 and the effectiveness of the constraints on the loss convergence trajectory. Candidate architectures were randomly sampled from the search space, and their performance was evaluated. The ground-truth values were then compared against the predicted values to analyze their correlation. As shown in Figure 2, the proposed MFTC method effectively leverages the low-dimensional concentration of high-weight parameters through a reliable module partitioning strategy and a linear prediction mechanism. It accurately estimates final performance while significantly reducing computational overhead during search and training. The module-wise performance estimates based on linear prediction exhibit a very strong statistical correlation with actual fine-tuning results ($Spearman > 0.9$), indicating that the predictive model can closely reflect the behavior of the fully trained model. Prediction errors remain consistently low, confirming that reliable estimates can be obtained via lightweight probing without incurring the expensive cost of full training. Furthermore, the loss convergence trajectories of the two models in Figures 2 (b) and (d) align closely, demonstrating that the modules extracted through DSCA exhibit stable convergence behavior during updates rather than resembling random noise. The method also shows strong generalization capability across models and tasks, with highly consistent prediction performance on both classification models (e.g., BERT) and generative models (e.g., GPT-2), proving that its effectiveness is not limited to specific architectures or task types.

**r × k ablation experiment:**Appendix E.2 study results indicate that model performance initially improves with increasing embedding dimension $r$, then gradually plateaus. When $r \geq 16$, it enters a region of diminishing returns, while computational resource consumption increases approximately linearly with $r$. Balancing performance and efficiency, $r$ is set to 16 by default. Excessively small values of cluster number $k$ lead to over-merged modules—although cross-block gradient similarity remains high, the model's expressive capacity becomes constrained.Conversely,overly large $k$ values result in fragmented cluster structures and amplify noise interference. Experimental results demonstrate that setting $k$ within the range [8, 12] yields optimal and robust performance across most tasks.

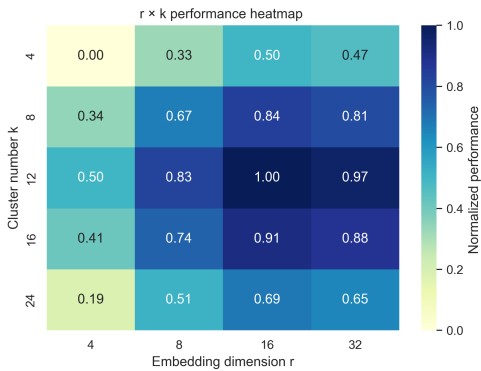

Figure 1: Interaction effect: $r \times k$ heatmap

Analysis of Heatmap 1 reveals a broad flat high-performance region around ($r = 16, k = 12$), indicating strong robustness within this parameter neighborhood.

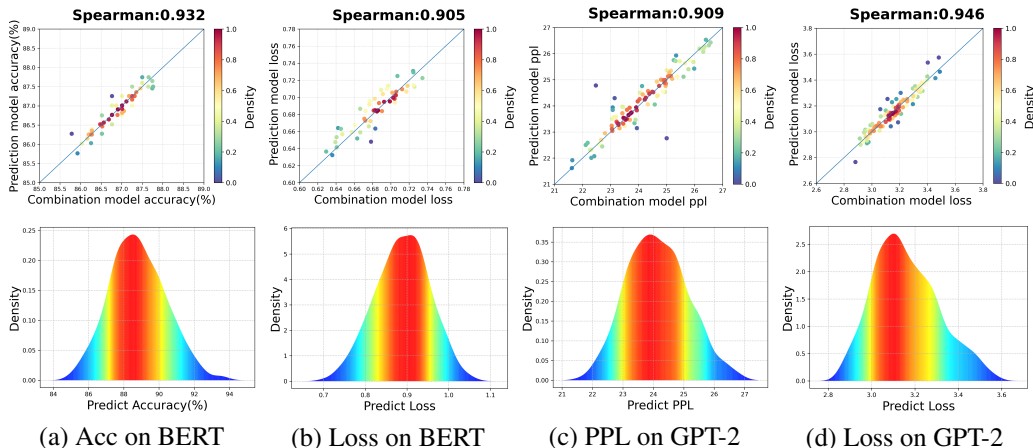

(a) Acc on BERT  (b) Loss on BERT  (c) PPL on GPT-2  (d) Loss on GPT-2

Figure 2: The correlation and density distribution between the performance of the linear predictive candidate models and the true training model performance are illustrated.(a) and (b) display the correlation and density distribution of accuracy and Loss for the linear function predictions of the candidate BERT models compared to the true values.(c) and (d) present the correlation and density distribution of the predicted values and true values for the GPT-2 model's PPL and Loss.

Table 1: Accuracy Comparison: MFTC vs. SOTA baselines in Fine-tuning of the BERT model.

| Methods | Update Parameter Quantity(M) | MRPC | | RTE | | SST-2 | |
|---|---|---|---|---|---|---|---|
| | | Avg. | △. | Avg. | △. | Avg. | △. |
| Full tuning | 113.25 | 89.79 | ±0.28 | 68.23 | ±0.79 | 89.22 | ±0.38 |
| Adapter | 0.96 | 90.38 | ±0.57 | 68.56 | ±1.06 | 88.31 | ±2.37 |
| AutoPEFT | 0.99 | 91.01 | ±0.38 | 69.31 | ±0.57 | 88.87 | ±1.03 |
| KD | 67.95 | 91.20 | ± 0.12 | 67.59 | ± 0.61 | 86.60 | ± 0.52 |
| SAM | 17.19 | 89.85 | ±0.17 | 69.19 | ±0.32 | 83.33 | ±2.32 |
| BitFit | 0.09 | 88.96 | ±1.31 | 65.04 | ±0.77 | 87.45 | ±0.99 |
| AutoLoRA | 0.27 | 89.87 | ±0.16 | 68.34 | ±0.87 | 89.32 | ±1.42 |
| MFTC | 1.77 | **93.38** | ± 0.15 | **71.48** | ± 0.29 | **90.37** | ± 0.81 |

## 4.3 MFTC FOR BERT

In the BERT experiments, we configured 12 nodes, each containing 16 dimensions per candidate module across 12 candidate modules. The fine-tuning hyperparameters were consistent with those specified in the official BERT documentation. As shown in Table 1, MFTC significantly outperforms existing PEFT methods and full fine-tuning while maintaining a low parameter update budget. Moreover, the improvements generally exceed the fluctuation ranges indicated by standard deviation, demonstrating robust results. Although MFTC requires updating 1.77M parameters (approximately 1.5% of the total), which is slightly higher than some other PEFT methods, the performance gains far outweigh the additional cost by avoiding the expense of full training. MFTC proves particularly effective in tasks requiring fine semantic alignment and logical reasoning, which aligns with the observation that high-weight parameters are concentrated in specific layers (e.g., layers 8–11 in BERT). The performance improvement is relatively modest on sentence-level sentiment classification tasks (+1.05%). Parameter analysis suggests that BERT exhibits significant full-parameter coupling characteristics in such tasks, with global features being more dispersed, which limits the ability of modular update strategies to effectively capture global semantic relationships.

## 4.4 EXTENDED EXPERIMENT

As shown in Table 2, the experimental results demonstrate that MFTC achieves superior performance over mainstream parameter-efficient fine-tuning methods across both 1.5B-parameter models and larger 7B mainstream models. These findings indicate that even on medium-scale models, the modular selection and updating of high-magnitude parameters can effectively reduce computational redundancy and mitigate overfitting. For larger models, task-specific parameters exhibit a more pronounced concentration trend, enabling MFTC's dynamic modular strategy to more precisely capture critical parameter subspaces. In contrast to traditional LoRA-series methods, MFTC not only achieves higher accuracy but also demonstrates more stable performance gains across different model scales, highlighting the method's enhanced generalizability and robustness.

Table 2: Comparison of fine-tuning performance across multiple magnitude models

| Method | GPT-2-1.5B | Llama-3.1-7B | Qwen-2.5-7B |
|--------|-----------|--------------|-------------|
| QLora | 23.68 | 73.98 | 78.66 |
| AdaLora | 24.09 | 73.69 | 78.35 |
| Pissa | 21.93 | 72.49 | 79.97 |
| MFTC | 20.52 | 75.97 | 80.93 |

Table 3: Comparison of energy consumption for different fine-tuning methods of BERT

| | Methods | Update Parameter(M) | GPU Time(s) |
|------|---------|---------------------|-------------|
| | FT | 113.25 | 1051.78±52.19 |
| | Adapter | 0.96 | 571.54±25.84 |
| | AutoPEFT | 0.99 | 566.26±41.47 |
| BERT | KD | 67.95 | 893.14±135.75 |
| | SAM | 17.19 | 677.38±109.43 |
| | BitFit | 0.09 | 505.19±61.28 |
| | AutoLoRA | 0.27 | 535.99±32.42 |
| | MFTC | 1.77 | 579.08±18.86 |

We conducted a systematic evaluation of the parameter efficiency and training overhead of classical BERT fine-tuning methods on a computational platform equipped with dual NVIDIA GeForce RTX 4090 GPUs. As shown in Table 3, the MFTC method demonstrates marked advantages in balancing performance and efficiency: it reduces training time by approximately 45% compared to full parameter fine-tuning, with trainable parameters amounting to only about $\frac{1}{64}$ of the latter. The average fine-tuning time for BERT using MFTC is 579 seconds, which is within the same order of magnitude as mainstream PEFT methods, yet MFTC achieves clearly superior performance across multiple tasks(see Table 1). These experimental results confirm that MFTC delivers stable and significant performance improvements with minimal computational overhead, demonstrating strong practical utility and deployment value.

## 5 CONCLUSION

This paper addresses the structural and efficiency bottlenecks in cross-task fine-tuning by systematically proposing and validating a technical pathway centered on "low-dimensional distribution of high-magnitude parameters" and "dynamic modular fine-tuning". Based on the properties of principal component predominance, approximate sparsity, and local smoothness, we establish a first-order approximation theory that models gradient updates within a low-dimensional subspace, and demonstrate that this subspace undergoes only slow drift during training—providing a verifiable theoretical foundation for our method. We further propose a Dynamic Spectral Clustering Algorithm and a lightweight modular search strategy, enabling stable tracking of the effective subspace and efficient combinatorial module selection with favorable time and space complexity scalability. Experiments across multiple task scenarios—including classification, matching, regression, and language modeling—show that the proposed method outperforms or matches mainstream baselines under comparable or lower parameter and computational budgets. Key diagnostic metrics further corroborate the theoretical predictions, including concentrated principal component energy, bounded update approximation error, and high module stability. Although the current approach faces limitations under strongly coupled tasks or abrupt distribution shifts, we suggest that mechanisms such as adaptive dimensionality adjustment, nonlinear embedding, and multi-objective search could extend its applicability. In summary, this work systematically demonstrates—from both theoretical and methodological perspectives—the effectiveness and practicality of the "low-dimensional structure hypothesis + dynamic clustering + lightweight search" pipeline. It offers an operable, verifiable, and scalable paradigm for modular fine-tuning and efficient cross-task transfer, thereby establishing a solid foundation for future research.

# 6 ETHICS STATEMENT

This work adheres to ethical research standards in data collection, model training, and evaluation. All datasets used in this study are publicly available research datasets (e.g.,BLUE,WikiText-2,Alpaca-20k,MMLU, which were collected and released under their respective licenses. No private or personally identifiable information (PII) was used.

# 7 REPRODUCIBILITY STATEMENT

We are committed to ensuring the reproducibility of our results. Comprehensive implementation details, including training data and hyperparameter settings, are provided in Appendices D and E. To further facilitate reproducibility, we will release the training code, datasets, and evaluation scripts upon publication.

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

# A  THEORETICAL DETAILS AND PROOFS

## A.1  CONCENTRATION AND RENEWAL APPROXIMATION OF PRINCIPAL COMPONENT ENERGY

(a) $\Sigma = U\Lambda U^T, \quad \Lambda = \mathrm{diag}(\lambda_i), \quad \Pi_r = U_r U_r^T.$

$$\mathbb{E}[\|\Pi_r g\|^2] = \mathrm{tr}(\Pi_r \Sigma) = \sum_{i=1}^{r} \lambda_i \geq (1-\varepsilon) \sum_{i=1}^{d} \lambda_i = (1-\varepsilon)\mathbb{E}[\|g\|^2].$$

(b) $\Delta\theta = -\eta G(\theta), \quad G(\theta) = \nabla L(\theta) + R(\theta), \quad \mathbb{E}[\|R\|^2] \leq C\eta B.$

$$\mathbb{E}[\|\Delta\theta - \Pi_r \Delta\theta\|^2] = \eta^2 \mathbb{E}[\|(I - \Pi_r)G\|^2]$$
$$\leq 2\eta^2 \mathbb{E}[\|(I - \Pi_r)\nabla L\|^2] + 2\eta^2 \mathbb{E}[\|R\|^2]$$
$$\leq 2\eta^2 \varepsilon \mathbb{E}[\|\nabla L\|^2] + O(\eta^3 B).$$

Also, $\mathbb{E}[\|\Delta\theta\|^2] = \eta^2 \mathbb{E}[\|G\|^2] = \eta^2 \left( \mathbb{E}[\|\nabla L\|^2] + O(\eta^2 B) \right).$

Normalizing yields the conclusion.

*Note: The spectral bound of the Hessian is not used; if the empirical Fisher is used instead of $\Sigma$, the form remains unchanged.*

## A.2  APPROXIMATE SPARSITY AND LOW-RANK CONSISTENCY

Let $h = B^T g$, with $\|h - h_s\|_2 \leq \varepsilon_s \|h\|_2$ and $\kappa(B)$ bounded. Then

$$\|g - B h_s\|_2 \leq \|B\|_2 \|h - h_s\|_2$$
$$\leq \kappa(B)^2 \varepsilon_s \|h\|_2$$
$$\leq \kappa(B)^4 \varepsilon_s \|g\|_2.$$

Extending to $r = O(s \log d)$ via greedy/randomized selection yields $\Pi_r$ such that

$$\|g - \Pi_r g\|_2 \leq C\kappa(B)^4 \varepsilon_s \|g\|_2,$$

where $C$ is a constant determined by the selection strategy.

## A.3  LOW-DIMENSIONAL CONSTRAINTS ON FRAGMENTED TRAJECTORIES

For time steps $t$ and $t+1$ within the same segment, we have $\|\Pi_r^{(t+1)} - \Pi_r^{(t)}\|_2 \leq \delta$. By Theorem 1(b):

$$\mathbb{E}[\|(I - \Pi_r^{(t)})\Delta\theta_t\|^2] \leq \varepsilon \mathbb{E}[\|\Delta\theta_t\|^2] + O(\eta^2 B).$$

Applying the triangle inequality:

$$\|(I - \Pi_r^{(t+1)})\Delta\theta_t\|_2 \leq \|(I - \Pi_r^{(t)})\Delta\theta_t\|_2 + \|(\Pi_r^{(t)} - \Pi_r^{(t+1)})\Delta\theta_t\|_2$$
$$\leq \sqrt{\varepsilon \|\Delta\theta_t\|^2 + O(\eta^2 B)} + \delta \|\Delta\theta_t\|_2$$
$$\leq \sqrt{\varepsilon} \|\Delta\theta_t\|_2 + \delta \|\Delta\theta_t\|_2 + O(\eta\sqrt{B}).$$

Taking expectations yields the conclusion.

## A.4  DISCUSSION ON THE UNDERLYING ASSUMPTIONS

The theoretical framework and analysis presented in this work do not rely on the strong assumption of local strong convexity. Although local strong convexity is often used in traditional optimization theory as a sufficient condition to guarantee algorithmic convergence, it is not a necessary condition for the validity of our approach. Our analysis demonstrates that under a more general non-convex optimization landscape, the methods of DSCA and modularization remain effective as long as two key properties hold: first, the gradient covariance matrix $\Sigma$ exhibits significant spectral dominance, meaning a low-dimensional subspace exists that captures the vast majority of its energy (guaranteed by $\kappa(r) \geq 1 - \varepsilon$); and second, this dominant subspace exhibits a property of slow drift over time

(monitored by the sequential subspace projection difference $|\mathbf{\Pi}_r^{(t+\Delta t)} - \mathbf{\Pi}_r^{(t)}|_2$). This implies that even for complex non-convex problems, effective and stable optimization can be achieved by tracking this slowly evolving principal component subspace. Furthermore, our proof strategy deliberately avoids using sophisticated differential geometry tools, such as the Nash Embedding Theorem, to prove the existence of a low-dimensional structure. Instead, it relies on spectral properties that can be directly estimated and validated from data (gradients). This choice makes the entire theoretical framework more operational, verifiable, and tightly integrated with the actual computational process of the algorithm.

## B  STABILITY AND DISTURBANCE ANALYSIS OF DSCA

### B.1  LAPLACE PERTURBATION BOUNDARY

Given the weight update rule:
$$W(t) = \alpha W(t-1) + (1-\alpha)S(t),$$
and the assumption that $\|S(t) - S(t-1)\| \le \varepsilon_s$ with minimum degree $d_{\min} > 0$.

First, we bound the change in weights:
$$\|W(t) - W(t-1)\| = \|[\alpha W(t-1) + (1-\alpha)S(t)] - W(t-1)\|$$
$$= \|(1-\alpha)(S(t) - W(t-1))\|$$
$$\le (1-\alpha)\|S(t) - W(t-1)\|.$$
Using the assumption $\|S(t) - S(t-1)\| \le \varepsilon_s$, we obtain:
$$\|W(t) - W(t-1)\| \le (1-\alpha)\varepsilon_s.$$

Now consider the symmetric normalized Laplacian $L_{\text{sym}} = I - D^{-\frac{1}{2}}WD^{-\frac{1}{2}}$. Using the bound $\|D^{-\frac{1}{2}}\|_2 \le 1/\sqrt{d_{\min}}$ and standard matrix perturbation theory for normalized Laplacians:
$$\|L_{\text{sym}}(t) - L_{\text{sym}}(t-1)\|_2 \le C \cdot \|D^{-\frac{1}{2}}\|_2^2 \cdot \|W(t) - W(t-1)\|$$
$$\le C \cdot \frac{1}{d_{\min}} \cdot (1-\alpha)\varepsilon_s,$$
where $C$ is a constant related to the bounds on normalization multipliers.

### B.2  FEATURE SUBSPACE PERTURBATION AND CLUSTERING STABILITY

Let $L_{\text{sym}}(t)$ and $L_{\text{sym}}(t-1)$ have spectral gap $\gamma > 0$ for their leading $r$-dimensional eigenspaces. By the Davis-Kahan $\sin\Theta$ theorem:
$$\|\sin\Theta(U_r^{(t)}, U_r^{(t-1)})\|_2 \le \frac{\|L_{\text{sym}}(t) - L_{\text{sym}}(t-1)\|_2}{\gamma}.$$

From our previous result on Laplacian perturbation:
$$\|L_{\text{sym}}(t) - L_{\text{sym}}(t-1)\|_2 \le C\frac{(1-\alpha)\varepsilon_s}{\sqrt{d_{\min}}}.$$

Combining these, the row-wise perturbation of the eigenvectors is bounded by:
$$\max_j \|u_j^{(t)} - u_j^{(t-1)}\|_2 \le \varepsilon_u \propto \frac{(1-\alpha)\varepsilon_s}{\gamma\sqrt{d_{\min}}}.$$

For k-means clustering applied to the eigenvector rows, the cluster centers satisfy:
$$\|m_i^{(t)} - m_i^{(t-1)}\|_2 \le \frac{1}{|C_i|}\sum_{j\in C_i}\|u_j^{(t)} - u_j^{(t-1)}\|_2$$
$$\le \max_j \|u_j^{(t)} - u_j^{(t-1)}\|_2$$
$$\le \varepsilon_u.$$

This establishes that cluster center movement is bounded by eigenvector perturbation, completing the proof.

# C LINEAR PREDICTION AND SEARCH CONVERGENCE

## C.1 LINEAR APPROXIMATION ERROR

Assume the true accuracy $\mathrm{Acc}(N)$ is given by

$$\mathrm{Acc}(N) = \mathrm{Acc}(N_e) + \sum_i \Delta A_{i,o(i)} + \epsilon_{\mathrm{int}},$$

where the interaction term $\epsilon_{\mathrm{int}}$ satisfies $|\epsilon_{\mathrm{int}}| \le \beta G(s)$, and $G(s)$ is monotonically non-increasing with the combination scale $s$ (marginal interaction decay). The mean absolute error (MAE) of estimating $\Delta A$ using a validation set is $\varepsilon_{\mathrm{est}}$. Then, the prediction error is bounded by $\varepsilon_{\mathrm{est}} + \beta G(s)$.

## C.2 APPROXIMATE SUBMODULARITY AND GREEDY GUARANTEE

Regarding approximate submodularity and the guarantee of the greedy algorithm: If for any sets $A \subseteq B$ and an additional module $b$, we have

$$F(A \cup \{b\}) - F(A) \ge F(B \cup \{b\}) - F(B) - \delta_s,$$

where $\delta_s$ decreases as $|B|$ increases (approximate submodularity), then the greedy algorithm after $k$ rounds satisfies

$$F(S_k) \ge (1 - 1/e)F(S^\star) - \sum_{t=1}^{k} \delta_t.$$

In practice, by (i) limiting the number of new modules added per step, and (ii) setting mutual exclusion or coupling penalties for nodes with strong interactions, we can reduce $\delta_t$.

## C.3 IMPLEMENTATION AND SOLUTION

This study employs a hybrid heuristic algorithm that combines a sequential node-wise greedy strategy with global 2-opt local search to solve the problem. During the stepwise node expansion phase, the algorithm iteratively selects the current optimal node based on a greedy principle to rapidly construct an initial feasible solution. A global 2-opt operation is then introduced to refine the path through pairwise edge exchanges, thereby enhancing solution quality and effectively avoiding local optima. To handle uncertain parameters in the model, a confidence lower bound criterion is adopted, which uses $\Delta A - \tau \sigma$ as the evaluation metric for uncertainty—where $\sigma$ denotes the empirical variance and $\tau$ is the confidence coefficient—transforming the uncertain problem into a tractable deterministic form. Furthermore, for small-scale instances, a relaxed integer linear programming (ILP) approach is applied to compute a lower bound, which helps assess the gap between the current solution and the optimal upper bound, thereby providing theoretical guarantees for the algorithm's performance. The proposed method achieves a favorable balance between computational efficiency and solution quality, combining the rapid convergence of heuristic search with the rigor of optimization theory.

# D MODULE GRANULARITY, SAMPLING, AND EXPANSION

## D.1 CHUNKING STRATEGY

A hierarchical blocking strategy is adopted for large-scale Transformer models, involving structured partitioning of parameter matrices within each neural network layer. Specifically, the six core weight matrices in each layer—namely, the query, key, value, and output projection matrices $(W_q, W_k, W_v, W_o)$, along with the two linear transformation matrices in the feed-forward network $(W_1, W_2)$—are partitioned into smaller memory blocks. The granularity of blocking can be selected among three modes according to computational and memory requirements: head-level blocking, column-level blocking, or fixed-width blocking (e.g., with $d_b = 16$ or $32$). In particular, for the embedding layer and LayerNorm modules, due to their unique parameter structures or data access patterns, a skipping strategy or coarse-grained aggregation may be applied to reduce memory overhead while maintaining computational efficiency. This blocking mechanism effectively mitigates memory fragmentation and improves hardware cache utilization, thereby facilitating the deployment and inference of larger models in resource-constrained environments.

## D.2   CHUNK GRANULARITY

The choice of blocking granularity inherently represents a trade-off between computational efficiency and representational capacity. Coarse-grained blocking (e.g., layer-level or head-level) enhances runtime speed by reducing both the number of blocks (smaller m) and computational overhead (smaller $E$), thereby improving memory access locality and facilitating large-scale parallel processing. However, this is achieved at the expense of reduced model expressiveness. Medium-grained blocking (e.g., block-level with a fixed width such as $d_b = 16$ or 32) strikes a balance between memory efficiency and model capacity, making it suitable for most mid-sized models. In contrast, fine-grained blocking (e.g., column-level) offers the highest expressive power and parameter flexibility, yet it significantly increases the number of blocks (larger m) as well as computational and storage costs. It is therefore recommended only for smaller models or scenarios with sufficient data sampling and abundant training resources. This hierarchical granularity system provides model developers with a tunable dimension that can be optimized according to hardware constraints and task requirements.

## D.3   DIAGNOSIS AND ADJUSTMENT STRATEGY

During model training based on dynamic memory and computational optimization, the system must adapt its execution strategy according to real-time diagnostic metrics. If the recorded memory utilization efficiency metric $\kappa(r)$ remains consistently low, it indicates inefficiency in the current memory allocation strategy. Three mitigation approaches can be employed: appropriately increasing the retention ratio parameter $r$, adjusting the blocking granularity to modify memory access patterns, or switching to parameter-efficient fine-tuning methods such as LoRA. If certain modules exhibit instability during training—manifested as low Normalized Mutual Information (NMI) or Adjusted Rand Index (ARI), or high fluctuation amplitude $\delta$ in latent representations—the constraint strength on these modules should be enhanced. This can be achieved by increasing the regularization coefficient $\alpha$, extending the sliding window length for gradient statistics, or introducing noise-reduction mechanisms—including gradient smoothing and Exponential Moving Average (EMA)—to improve training stability and representational consistency. This diagnostic and adaptive strategy adjustment mechanism forms a closed-loop optimization framework, significantly enhancing the robustness and resource efficiency of large-scale model training.

# E   ABLATION EXPERIMENT ON EMBEDDING DIMENSION R AND CLUSTER NUMBER K

## E.1   PRINCIPLE OF $\alpha$ VALUE SELECTION

When determining the value of the smoothing factor $\alpha$, the following principles should be observed: In the presence of high-noise gradients during training, a higher $\alpha$ value (e.g., 0.9–0.98) is recommended to enhance gradient smoothing, suppress anomalous fluctuations, and improve training stability. For scenarios involving rapidly shifting task distributions or requiring high adaptability, a moderate $\alpha$ value (e.g., 0.8–0.9) is advised to maintain a certain degree of smoothing while increasing responsiveness to dynamic environments. In general settings, a default value between 0.9 and 0.95 is adopted to balance robustness and adaptability in the optimization process. This principled approach to hyperparameter tuning aims to optimize both the convergence behavior and generalization performance of model training.

## E.2   ABLATION EXPERIMENT ON EMBEDDING DIMENSION R AND CLUSTER NUMBER K

Experimental results across diverse tasks demonstrate that model performance initially increases with the embedding dimension $r$, then progressively saturates; when $r \geq 16$, further improvements enter a phase of diminishing marginal returns. Meanwhile, both computational and storage costs exhibit an approximately linear growth with respect to $r$. Based on this comprehensive trade-off between performance and resource consumption, the default embedding dimension is set to $r = 16$. This empirical conclusion aligns with the theoretical analysis presented in Section 3: when the memory utilization efficiency metric $\kappa(r) \geq 0.95$, the error induced by first-order approximation

updates can be consistently confined within 10%–15%, ensuring both theoretical reliability and computational effectiveness of the proposed method.

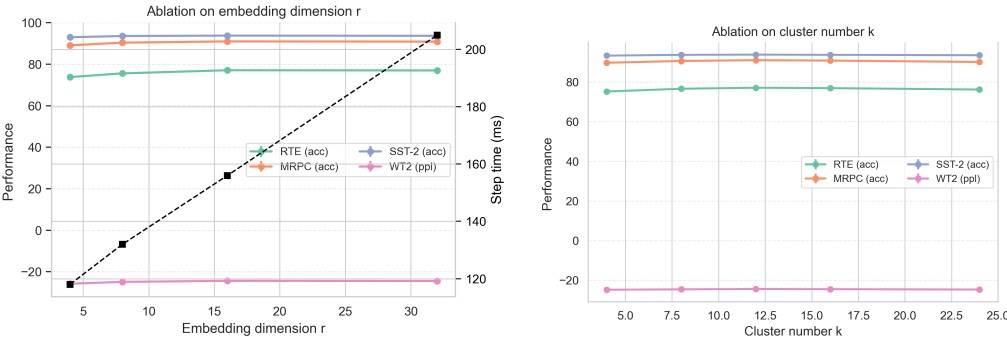

Figure 3: Performance curve of r and k

On the other hand, the choice of the number of clusters $k$ significantly influences model behavior: an excessively small $k$ leads to over-consolidation of functional modules, which may preserve high cross-block gradient similarity but compromises the model's expressive power; conversely, an excessively large $k$ tends to cause cluster fragmentation and amplifies the effect of noise. Extensive experimental results indicate that setting $k$ within the range [8, 12] achieves optimal and robust performance across the majority of tasks.

# F   USE OF LLMS

Yes. We used LLMs solely to assist in language polishing and improving readability. All technical content, experiments, and analyses were conducted by the authors.

