# OpenReview forum: "Modular Fine-Tuning of Clustering:Directional Updating of Weight Parameters for PLMs"
_ICLR.cc/2026/Conference — Submitted to ICLR 2026_

### Official Review · Reviewer_M64r · 2025-10-31

**Soundness:** 2
**Presentation:** 1
**Contribution:** 2
**Rating:** 2
**Confidence:** 4

**Summary:**

This paper proposes MFTC (Modular Fine-Tuning via Clustering), a parameter-efficient fine-tuning framework for large language models. It is based on the observation that high-magnitude parameters in PLMs concentrate within a low-dimensional subspace that dominates gradient updates. The method employs a Dynamic Spectral Clustering Algorithm (DSCA) to identify and group these influential parameters into functional modules, which are selectively fine-tuned. A lightweight linear predictor with neural architecture search then selects the optimal module combinations without full retraining.

Experiments on BERT, GPT-2, Llama-3.1, and Qwen-2.5 demonstrate that MFTC achieves better accuracy and 40–50% lower energy cost compared to LoRA, Adapter, and other PEFT methods, while updating only ~1.5% of parameters.

**Strengths:**

* Strong theoretical grounding on low-dimensional gradient concentration.
* Innovative dynamic modular clustering and NAS-based selection.
* Superior performance–efficiency balance with broad scalability and robustness.

**Weaknesses:**

Clarity and Reproducibility
The paper does not report the hyperparameters used for fine-tuning. Please include them, as the variance of RTE results is usually higher than SST-2, which may affect the interpretation of results.

The definitions of symbols in Equations (1)–(2) and (6)–(7) are missing. Please clearly define θ1, θ2, and μ, and use the proper LaTeX notation \mathrm{} for functions like Acc and Loss.

Overall, the notation and mathematical presentation require clarification; currently, the equations are difficult to follow without explicit symbol declarations. Piling up mathematical equations will not make your paper look more impressive; it will only undermine its clarity and overall presentation.


Writing and Presentation Quality
The manuscript contains numerous spacing and formatting issues. For example:
Line 142: a missing space after “if there exists a projection”;
Missing spaces around “(a)” and “(b)” after “such that”;
Line 22: missing space between “.” and “Building.”
In Line 832, “parameter-efficient fine-tuning” should be abbreviated as PEFT.
The paper lacks figures or visual explanations in the Introduction and Method sections, making it difficult to read. Adding illustrative diagrams would significantly improve readability and conceptual clarity. This paper requires significant refinement.

**Questions:**

Accuracy and Completeness of Tables
In Table 2, several method names are incorrect and even without citations: QLora should be OLoRA, AdaLora should be AdaLoRA, and Pissa should be PiSSA. Please correct these and cite the corresponding original papers. Additionally, please confirm whether “Llama-3.1-7B” refers to “Llama-3.1-8B,” as this appears inconsistent.
The choice of baselines in Table 2 is questionable, as QLoRA, AdaLoRA, and PiSSA are derived from different methodological perspectives. You should clearly define the specific problem your work aims to address within the PEFT framework and select baselines that target the same problem, rather than including methods solely because they are “popular.”

Experimental Design and Benchmark Coverage
The paper only reports results on MRPC, RTE, and SST-2. Please justify this limited selection and explain why the full GLUE benchmark or more diverse reasoning datasets (e.g., ARC-c, ARC-e, PIQA, SIQA, OBQA, HellaSwag, Winogrande, BoolQ) were not included.
Section 4.4 (“Extended Experiment”) lacks clarity regarding the specific tasks and metrics presented. Please specify what each score represents.
More NLG-oriented experiments should be conducted to demonstrate broader applicability.
An ablation study and analysis under different hyperparameter settings (e.g., number of nodes, rank dimensions) are necessary to support the claimed contributions.

---

### Official Review · Reviewer_ECZf · 2025-11-02

**Soundness:** 2
**Presentation:** 1
**Contribution:** 2
**Rating:** 2
**Confidence:** 4

**Summary:**

This paper proposes Modular Fine-Tuning of Clustering (MFTC), a new approach for parameter-efficient fine-tuning of pretrained language models (PLMs). The
method builds on the observation that high-magnitude gradients and parameter weights tend to lie within a low-dimensional subspace. MFTC introduces a
Dynamic Spectral Clustering Algorithm (DSCA) to identify and group highly correlated parameter subsets into modules and then uses Neural Architecture Search
(NAS) to select and optimize combinations of these modules for downstream tasks. The approach effectively combines ideas from parameter subspace analysis,
clustering, and architecture search to improve efficiency and adaptability in fine-tuning. Experimental results on BERT, GPT-2, LLaMA, and Qwen show that MFTC
achieves competitive or superior performance compared to strong baselines such as LoRA, Adapter, and AutoPEFT, while maintaining low computational and
parameter overhead.

**Strengths:**

A substantive assessment of the strengths of the paper, touching on each of the following
dimensions: originality, quality, clarity, and significance. We encourage reviewers to be broad in
their definitions of originality and significance. For example, originality may arise from a new
definition or problem formulation, creative combinations of existing ideas, application to a new
domain, or removing limitations from prior results.

1. The paper proposes an interesting cross-disciplinary idea that merges spectral clustering, low-dimensional subspace theory, and neural architecture search for parameter optimization. The notion of modularizing parameters based on gradient concentration patterns offers a theoretically grounded and biologically inspired perspective.

2. Experiments across diverse tasks and model scales are well structured. The results convincingly demonstrate that MFTC outperforms existing BERT fine-tuning baselines in both accuracy and training efficiency.

3. The paper presents clear motivation for its design choices, such as the use of gradient aggregation and the linear prediction model, supported by both
intuitive reasoning and empirical evidence.

**Weaknesses:**

A substantive assessment of the weaknesses of the paper. Focus on constructive and actionable
insights on how the work could improve towards its stated goals. Be specific, avoid generic
remarks. For example, if you believe the contribution lacks novelty, provide references and an
explanation as evidence; if you believe experiments are insufficient, explain why and exactly what is
missing, etc.

1. Several notations are ambiguous or introduced without sufficient explanation. For example, in Eq. (7), the meaning of b_{ij}^O​ (the selection indicator
variable) is only clarified later in Eq. (8); defining it earlier would improve readability. In addition, the use of b_{ij}^A​ and b_{ij}^L​ ​could be confusing
since they share similar notation with the node variable node b_ij.

2. The paper selects the smallest r eigenvalues for spectral embedding, but the rationale for this choice is not well justified. It would strengthen the
argument to explicitly connect this selection to gradient concentration theory or provide empirical evidence (e.g., spectral energy distribution analysis)
supporting this decision.

3. There is no comparison of resource usage with other methods in Table 2. Furthermore, it remains unclear whether the “lightweight fine-tuning” process
described in Eq. (7) introduces additional computational or memory overhead.

4. Are there experimental results confirming that the proposed method faces limitations under abrupt distribution shifts? How do other parameter-efficient
fine-tuning methods behave under the same conditions?

5. Beyond evaluations on the MMLU benchmark, does the method maintain strong performance on other types of downstream tasks or domains?

**Questions:**

See weakness

---

### Official Review · Reviewer_z7H9 · 2025-11-02

**Soundness:** 2
**Presentation:** 3
**Contribution:** 2
**Rating:** 6
**Confidence:** 4

**Summary:**

This paper proposes a dynamic and efficient fine-tuning method called Modular Fine-Tuning of Clustering (MFTC).
The approach identifies and updates task-relevant parameter subsets through three steps:
1. The model is divided into multiple structural nodes, and a sparse similarity matrix is constructed from gradient similarity using the Top-$k$ (or Top-$p$) strategy.
2. A Dynamic Spectral Clustering Algorithm (DSCA) clusters parameters with correlated gradient patterns into functional modules.
3. A linear prediction–based module search then selects one module per node to fine-tune, optimizing accuracy under a loss constraint without retraining all candidates.

Theoretical analysis verifies that gradient updates lie in a low-dimensional subspace, and experiments on BERT, GPT-2, Llama-3.1, and Qwen-2.5 show that MFTC achieves comparable efficiency to recent PEFT methods with better performance.

**Strengths:**

1. MFTC achieves notable improvements on BERT, Llama-3.1-7B, and Qwen-2.5-7B, reducing GPU time by about 45% compared with full-parameter fine-tuning while maintaining comparable efficiency to recent PEFT methods.
2. This paper provides a thorough theoretical foundation and empirical analysis supporting MFTC’s reliability, effectiveness, and energy efficiency.

**Weaknesses:**

1. The LLM experiments rely solely on a single dataset (Alpaca-20k for training and MMLU for evaluation), which limits the generalizability of the results. Additional benchmarks (e.g., ARC, HellaSwag, TruthfulQA) would strengthen the empirical evidence.
2. Although most models demonstrate improvement, GPT-2-1.5B shows a noticeable performance drop in Table 2 without explanation. The paper should further analyze potential causes, such as model scale sensitivity or clustering instability.
3. (Minor) The two key hyperparameters, embedding dimension $r$ and cluster number $k$, are only ablated on BERT-large. The paper lacks broader investigation or guidance on selecting these parameters across different architectures or tasks.

[Improvement]

In Section 3.2.2, the $L_sym(t)$ should be $L_\mathrm{sym}(t)$.

**Questions:**

Section 3.2.4 mentions that $\alpha\in [0.85, 0.95]$ achieves a balance between stability and sensitivity; how was this range determined across different models? Does $\alpha$ need to be tuned per task, or can a fixed value generalize?

---

### Official Review · Reviewer_epjM · 2025-11-03

**Soundness:** 3
**Presentation:** 2
**Contribution:** 2
**Rating:** 2
**Confidence:** 4

**Summary:**

This paper introduces a new parameter efficient fine-tuning method. The main idea is to identify the representative modules by spectral clustering based on gradient. In detail, the method constructs a similarity graph from gradients, applied spectral clustering to group the modules, and select important modules per each cluster using a linear approximation and estimation.

It differs from prior PEFT methods because it discovers fine-tuning modules automatically from gradient dynamics, instead of hand-designing modules.

The authors evaluate their approach on multiple small-scale language models and several mid-size models, demonstrating competitive performance with significantly fewer trainable parameters.

**Strengths:**

The motivation of this paper is solid. Recent work has shown the redundancy in large model architectures, where different layers or blocks contribute unequally to final performance. The authors notice this and try to automatically find important modules and then perform parameter efficient fine-tuning on those modules.

The paper also provides theoretical support for the spectral clustering-based module selection process.

**Weaknesses:**

1. The experimental results need more clarification. Based on the results reported in [1], the full fine-tuning and AutoPEFT baselines appear weaker here (e.g., SST-2 ~92.57 in [1] vs. 89.22 reported). I may have missed some details, but the authors should explain the baseline setup more clearly to ensure a fair comparison. In Table 2, the paper does not specify whether the setting is 5-shot, 0-shot, or another setting, which makes it difficult to interpret the results. Alpca-20k is not a wide-used datasets for benchmark either.
2. Figure 1 suggests that the method can be sensitive to hyper-parameters.



[1] AUTOPEFT: Automatic Configuration Search for Parameter-Efficient Fine-Tuning

**Questions:**

- In line 189, gi is described as “the gradient vector or its low-dimensional sketch.” The use of “or” is unclear. Which form is actually used in the experiments? How do the authors handle the case where gradient vectors have different dimensions across modules?

- This paper includes a lot of modules, Besides for r and k, did the authors perform ablation experiments for each component? What is the performance if we randomly select one module from each cluster without using the linear approximation step? What is the performance if we only use linear approximation without clustering? The reviewer is confused about the necessity and contribution of each part of the pipeline.

- The method uses gradient similarity to construct the module relationship graph. Have the authors tried using other statistics, such as gradient variance or other stability measures, to compute similarity?

- For Table 3, does the reported time include the entire pipeline (clustering + linear approximation + fine-tuning), or only the fine-tuning after selected modules are identified?

---

### Meta-Review · Area_Chair_WgHs · 2026-01-06

**Summary:**

This paper proposes MFTC, a parameter-efficient fine-tuning approach that uses gradient-based graph construction + dynamic spectral clustering to form “modules,” then applies a lightweight predictor / NAS-style selection to choose modules to fine-tune. Reviews are split with the decision hinging less on the high-level idea and more on clarity/reproducibility and experimental credibility: unclear/possibly weak baseline setups, limited and nonstandard benchmarks for LLMs, missing resource accounting, and pervasive notation/presentation issues.
Modular Fine-Tuning of Clusteri…

**Reviewer Concerns:**

No author rebuttal/discussion content is visible in the provided snapshot, so I cannot confirm resolution of concerns.

**Reviewer Scores:**

Given the (apparent) absence of rebuttal/discussion and the strength of the core concerns, I expect little rating change.

---

### Decision · Program_Chairs · 2026-01-26

Reject